Metabarcoding monitoring analysis: the pros and cons of using co-extracted environmental DNA and RNA data to assess offshore oil production impacts on benthic communities

Laroche Olivier olar785@aucklanduni.ac.nz olli.laroche@gmail.com 1 2
Wood Susanna A. 2 3
Tremblay Louis A. 1 2
Lear Gavin 1
Ellis Joanne I. 4
Pochon Xavier 2 5
1 School of Biological Sciences, University of Auckland , Auckland , New Zealand
2 Environmental Technologies, Coastal and Freshwater Group, Cawthron Institute , Nelson , New Zealand
3 Environmental Research Institute, University of Waikato , Hamilton , New Zealand
4 Red Sea Research Centre, King Abdullah University of Science and Technology , Thuwal , Saudi Arabia
5 Institute of Marine Science, University of Auckland , Auckland , New Zealand
Reimer James
Electronic publication date: 2017 May 17
Publication date: 2017
Volume: 5
Electronic Location ID: e3347
Received 2017 Feb 1; Accepted 2017 Apr 22
Copyright: ©2017 Laroche et al.
Copyright year: 2017
Copyright holder: Laroche et al.
License: This is an open access article distributed under the terms of the Creative Commons Attribution License, which permits unrestricted use, distribution, reproduction and adaptation in any medium and for any purpose provided that it is properly attributed. For attribution, the original author(s), title, publication source (PeerJ) and either DOI or URL of the article must be cited.
License URL: https://creativecommons.org/licenses/by/4.0/

Keywords: Biomonitoring, Oil and gas activities, Benthic ecology, High-throughput sequencing, Bacteria (16S), Eukaryotes (18S), Method testing, eDNA, eRNA

Funding: Cawthron Institute Internal Investment Fund IIF #15955 Fonds de Recherche du Québec—Natures et Technologies ID#184395 This research was co-funded by the Cawthron Institute Internal Investment Fund (IIF #15955) and the “Fonds de Recherche du Québec—Natures et Technologies” as part of a doctoral research scholarship (grant ID#184395). The funders had no role in study design, data collection and analysis, decision to publish, or preparation of the manuscript.

==============================
Sequencing environmental DNA (eDNA) is increasingly being used as an alternative to traditional morphological-based identification to characterize biological assemblages and monitor anthropogenic impacts in marine environments. Most studies only assess eDNA which, compared to eRNA, can persist longer in the environment after cell death. Therefore, eRNA may provide a more immediate census of the environment due to its relatively weaker stability, leading some researchers to advocate for the use of eRNA as an additional, or perhaps superior proxy for portraying ecological changes. A variety of pre-treatment techniques for screening eDNA and eRNA derived operational taxonomic units (OTUs) have been employed prior to statistical analyses, including removing singleton taxa (i.e., OTUs found only once) and discarding those not present in both eDNA and eRNA datasets. In this study, we used bacterial (16S ribosomal RNA gene) and eukaryotic (18S ribosomal RNA gene) eDNA- and eRNA-derived data from benthic communities collected at increasing distances along a transect from an oil production platform (Taranaki, New Zealand). Macro-infauna (visual classification of benthic invertebrates) and physico-chemical data were analyzed in parallel. We tested the effect of removing singleton taxa, and removing taxa not present in the eDNA and eRNA libraries from the same environmental sample (trimmed by shared OTUs), by comparing the impact of the oil production platform on alpha- and beta-diversity of the eDNA/eRNA-based biological assemblages, and by correlating these to the morphologically identified macro-faunal communities and the physico-chemical data. When trimmed by singletons, presence/absence information from eRNA data represented the best proxy to detect changes on species diversity for both bacteria and eukaryotes. However, assessment of quantitative beta-diversity from read abundance information of bacteria eRNA did not, contrary to eDNA, reveal any impact from the oil production activity. Overall, the data appeared more robust when trimmed by shared OTUs, showing a greater effect of the platform on alpha- and beta-diversity. Trimming by shared OTUs likely removes taxa derived from legacy DNA and technical artefacts introduced through reverse transcriptase, polymerase-chain-reaction and sequencing. Findings from our scoping study suggest that metabarcoding-based biomonitoring surveys should, if funds, time and expertise allow, be assessed using both eDNA and eRNA products.

Introduction

Environmental DNA (eDNA) metabarcoding, defined here as the combined use of universal DNA barcodes and high-throughput sequencing (HTS) to characterize biological communities from genetic material collected from environmental samples (sediment, water, etc.), is increasingly being used to assess biodiversity and anthropogenic impacts in terrestrial (e.g., Yu et al., 2012; Taberlet et al., 2012; Ji et al., 2013; Beng et al., 2016) and aquatic environments (e.g., Carew et al., 2013; Chariton et al., 2014; Visco et al., 2015; Dowle et al., 2015; Pochon et al., 2015; Pawlowski et al., 2016a; Pawlowski et al., 2016b; Abad et al., 2016; Lanzén et al., 2016). Metabarcoding is a cost-effective method that can rapidly and simultaneously target multiple species to complement traditional ecosystem biomonitoring approaches (Valentini, Pompanon & Taberlet, 2009 Bourlat et al., 2013; Ji et al., 2013; Bohmann et al., 2014). It also allows microbiota that are difficult or impossible to identify morphologically to be characterized. Micro-organisms are at the bottom of the food chain, have shorter life cycles, and often have a higher diversity and abundance than macro-fauna. They may therefore be a better proxy for evaluating environmental impacts, and their use has been advocated for biomonitoring (Bourlat et al., 2013; Pawlowski et al., 2014; Pawlowski et al., 2016b; Pawlowski, Lejzerowicz & Esling, 2014; Lejzerowicz et al., 2015; Dowle et al., 2015; Lau et al., 2015).

Most metabarcoding monitoring studies use eDNA to characterize biological communities. A limited number have also evaluated data from co-extracted eRNA products (e.g., Pawlowski et al., 2014; Pawlowski et al., 2016a; Visco et al., 2015; Dowle et al., 2015; Pochon et al., 2015; Laroche et al., 2016). Because RNA deteriorates rapidly after cell death, eRNA likely provides a more accurate representation of viable communities (Mengoni et al., 2005). Blazewicz et al. (2013) suggested that the relative concentration of RNA in the environment provides a robust indication of the growth and adaptation potential of microbial communities. In general, when environmental metabarcoding studies have used both eDNA and eRNA they have found slightly stronger correlations between community data generated from eRNA and environmental conditions (Pawlowski et al., 2014; Pawlowski et al., 2016a; Visco et al., 2015; Dowle et al., 2015; Pochon et al., 2015; Laroche et al., 2016). However, working with eRNA adds additional complexities and expenses related to sampling and laboratory analysis including extra precaution during sampling, transport and storage to avoid degradation. A greater understanding of the potential benefits of using eDNA versus eRNA is therefore desirable prior to incorporating metabarcoding methods into environmental monitoring programs.

Previous studies assessing eDNA and eRNA extracted from the same environmental sample have noted differing diversities of operational taxonomic units (OTUs) recovered from assessment of either molecule. Greater taxon richness derived from eDNA data can be explained through the detection of DNA from dead organisms as well as extracellular DNA (free-floating or legacy DNA) that has bound to sediment or other particles. However, many studies report an important number of taxa only detected from eRNA (e.g., Pawlowski et al., 2014; Pawlowski et al., 2016a; Pochon et al., 2015; Laroche et al., 2016; Hu et al., 2016). These OTUs may be the result of PCR/sequencing artefacts or the detection of rare but very active taxa. To date, most studies, have dealt with this by only retaining the data of taxa present in both eDNA/eRNA datasets across the entire sample set (Pawlowski et al., 2014; Pawlowski et al., 2016a; Dowle et al., 2015; Pochon et al., 2015; Laroche et al., 2016), or within the same sites (Hu et al., 2016).

A further data treatment technique which is commonly applied to datasets, is the use of minimal DNA sequence read abundance thresholds for each unique taxon (i.e., data related to any particular OTU is discarded if insufficient numbers of that taxon are detected in the overall dataset, typically 2–10 reads). This is generally done to reduce sequencing artefacts (Hunt et al., 2013; Charvet, Vincent & Lovejoy, 2014; Visco et al., 2015). The methods applied to the pre-treatment of metabarcoding data (i.e., use of a minimum abundance removal threshold or dataset trimming based on shared-taxa abundances) can have important implications on the final resolution (number of available DNA sequence reads and/or taxa), sensitivity (smallest perceptible effect) and accuracy (confidence level) of the data used to assess anthropogenic impacts.

The aims of this study was to compare the use of co-extracted eDNA and eRNA products to assess the benthic environment around an offshore oil production platform, and to determine whether eDNA- or eRNA-derived taxa provide more accurate data for assessing shifts in bacterial and micro/meio-eukaryotic communities. We also evaluated the effects of two different trimming methodologies: one using a minimum read abundance threshold per taxon (i.e., ‘trimmed by singletons’), and one removing all taxa not simultaneously recovered from both eDNA and eRNA products of the same environmental sample (i.e., ‘trimmed by shared OTUs’). We hypothesized that: (1) bacterial and eukaryotic datasets with taxa trimmed by shared eDNA and eRNA products would yield more significant community shifts in response to the effect of the production platform from the removal of most irrelevant OTUs; and (2) that the use of eRNA-derived bacterial and eukaryotic taxa would show stronger correlations with environmental variables and macro-faunal communities than eDNA-derived taxa, as it mostly depicts the living community.

Figure 1 Site map indicating: (A) The study site situated in the Taranaki Bight (Maari facility, black circle); (B) The Floating Production Storage and Offloading (FPSO) platform.

Sampling stations are in full circle, with numbers corresponding to the distance to the platform along the major water flow axis. Modified from Johnston et al. (2014).

Material and Methods

Field sampling

The study area was located 80 km off the west coast of New Zealand’s North Island in the South Taranaki Bight (latitude: −39.97295°, longitude: 173.2997°; Fig. 1A). In this region, the seabed has a fairly uniform muddy texture with water depth ranging between 100 and 125 m. Biological samples (bacterial, micro-eukaryotic and macro-faunal assemblages) and physico-chemical data were collected along transects radiating to the East–West axis of one Floating Production Storage and Offloading (FPSO) platform. The FPSO has been in activity since 2009 and has been processing and storing the well fluids coming from a wellhead platform (WHP) situated 1.3 km North-West. The main discharges from the FPSO are production waters (formation water and additives) and operational discharges (sewage, deck drainage, etc.; (McConnell et al., 2014)). The FPSO site is characterized by a prevalence of anthropogenic debris (e.g., paint chips, construction material, etc.) and coarser, hard-packed and dark-coloured sediments in its vicinity (≤250 m), possibly indicating lower oxygenation level, elevated barium concentrations radiating away from the production water and drilling discharge point, and differences in the structure of macro-faunal assemblages on the east transect, between the near field station (≤250 m) and far-field stations (>250 m; Johnston, Elvines & Newton, 2015).

The sampling methodology was based on the Offshore Taranaki Environmental Monitoring Protocol (OTEMP; Johnston et al., 2014), which consisted of a distance-graded sample station allocation. Sampling stations were overlaid along the major flow axis with the FPSO located at the center (Fig. 1B). The East–West axis constitutes the main trajectory along which the deposition of drilling mud and cuttings occurs, with the strongest currents usually flowing in a westward direction at a mean speed of 0.12 ms−1 (McConnell et al., 2014). Six stations were sampled at approximately 250, 500 and 1,000 m from the FPSO.

Sediment samples were collected in triplicate at 6 stations for a total of 18 samples, on the 15th and 16th of March 2015 (Table S1). Samples were collected with a modified stainless steel double van Veen grab as described in Laroche et al. (2016). To avoid creating a bow wake effect on the seafloor, the grab sampler was deployed at a constant rate of 0.3 m/s. Recovery of the sampler was similarly processed to avoid displacing superficial sediments. Once retrieved, the top sediment layer was inspected and samples only taken when the surface was undisturbed. The full content of the first grab compartment was sieved (500 µm) and preserved in 70% ethanol for analysis of macro-faunal communities. Subsamples (2 g) of undisturbed surface sediment (approximately 1 cm depth) were collected from the second grab for eDNA/eRNA metabarcoding. These samples were placed in Life Guard™ Soil Preservation Solution (5 mL; MoBio, New York, NY, USA) to preserve microbial and environmental RNA in stasis, using disposable gloves and spatulas. Samples were stored on ice during transportation to the laboratory and kept frozen (−20 °C) until further processing. The second grab compartment was subsampled for physicochemical analyses of sediment as described in Johnston, Elvines & Newton (2015).

Laboratory analyses

Sample processing

The macro-fauna was morphologically identified to the lowest practical taxonomic level and enumerated at the Cawthron Institute (Nelson, New Zealand). Sediment texture and chemical analyses were also performed at Cawthron Institute, following the protocol described in Johnston et al. (2014). A summary of all analytical methods used and their detection limit for characterizing the physicochemical parameters of sediment is provided in supplementary information (Table S2 ).

Total DNA/RNA co-extractions were conducted using the PowerSoil™ Total RNA Isolation Kit and the DNA Elution Accessory Kit (MoBio, New York, NY, USA). Prior to cDNA synthesis, RNA samples were given two consecutive DNase treatments as described in Langlet et al. (2013). To verify the complete elimination of trace DNA, Polymerase Chain Reaction (PCR) amplifications of 50 cycles were performed using the bacterial forward primers S-D-Bact-0341-b-S-17: 5′-CCT ACG GGN GGC WGC AG-3′ and reverse primers S-D-Bact-0785-a-A-21: 5′-GAC TAC HVG GGT ATC TAA TCC-3′ (Klindworth et al., 2013), and the eukaryotic forward primers Uni18SF: 5′-AGG GCA AKY CTG GTG CCA GC-3′and reverse primers Uni18SR: 5′-GRC GGT ATC TRA TCG YCT T-3′ (Zhan et al., 2013). These primers produce amplicons of ca. 450 bp and were chosen for their ability to capture most bacterial and eukaryotic (especially protist) phyla. Complementary DNA (cDNA) synthesis from single-stranded RNA was performed using random hexamer primers and the SuperScript® III reverse transcriptase (Thermo Fisher Scientific Inc., Waltham, MA, USA), all according to the manufacturer’s manual.

PCR amplification and high-throughput sequencing

PCR amplifications were performed once for each biological replicate. Total bacterial 16S rRNA (V3 and V4 region) and total eukaryote 18S rRNA (V4 region) were amplified using PCR with the same primers descripted above. The primers were modified to include Illumina™ overhang adaptors forward: 5′-TCG TCG GCA GCG TCA GAT GTG TAT AAG AGA CAG-3′ and reverse: 5′- GTC TCG TGG GCT CGG AGA TGT GTA TAA GAG ACA G-3′, as described in Kozich et al. (2013). For bacteria, PCR consisted of 20 µL of AmpliTaq Gold® 360 PCR Master Mix (Life Technologies, Carlsbad, CA, USA), 8 µL of double-distilled water (ddH2O), 1 µL of each primer (10 µM), 5 µL of GC enhancer to increase yield, 2 µL (1 µg/ µL) of Bovine Serum Albumin (BSA), and 3 µL of template eDNA (1.58–3.96 ng/µL) or 1 µL of cDNA (10 ng/ µL). Reaction cycling conditions were: 94 °C for 3 min, followed by 30 cycles of 94 °C for 30 s, 52 °C for 30 s, 72 °C for 1 min, with a final extension step at 72 °C for 5 min. For eukaryotes, PCR reactions were performed in a reaction mixture containing 25 µL of AmpliTaq Gold® 360 PCR Master Mix (Life Technologies, Carlsbad, CA, USA), 13 µL of ddH2O, 1 µL of each primer (10 µM), 7 µL of GC enhancer, 3 µL of BSA, and 3 µL of template eDNA (1.58–3.96 ng/ µL) or 1.5 µL of eRNA (10 ng/ µL). Reaction cycling conditions were: 95 °C for 5 min, followed by 35 cycles of 94 °C for 30 s, 54 °C for 30 s, 72 °C for 45 s, with a final extension step at 72 °C for 7 min. All PCR amplifications included negative controls (no template samples) to check amplification of uncontaminated products. Amplicons were purified with Agencourt® AMPure® XP PCR Purification beads (Beverly, MA, USA) following the manufacturers’ instructions. Purified products were quantified using a Qubit® Fluorometer (Life Technologies, Carlsbad, CA, USA), and diluted to 3 ng/ µL using ddH2O. Each product was individually indexed using the Nextera™ DNA library Prep Kit (Illumina, San Diego, CA, USA), pooled into two libraries (eDNA and eRNA) and sequenced on a MiSeq™ Illumina platform using a 2 × 250 base pair paired-end protocol at the New Zealand Genomics Ltd. (Auckland, New Zealand). A sample containing 20 µL of ddH2O was used as a negative control for the multiplexing and sequencing steps in each library. Internal sequencing quality control was also ensured by including DNA samples from three previously isolated marine species (i.e., Sabella spallanzanii, Ciona savignyi and Perna perna (Pochon et al., 2013)) that were PCR amplified, purified and quantified following the eukaryote DNA protocol described above. These species were pooled at equimolar concentration and included in each sequencing run. The raw sequencing reads are publicly available in the Sequence Read Archive (SRA) under the accession numbers SRR5215546–SRR5215617. Additionally, the filtered sequence dataset is available for download on FigShare at the following link: https://figshare.com/articles/Laroche_et_al_2017_Maari/4600174.

Bioinformatics analysis

Using VSEARCH (Rognes et al., 2016), raw FASTQ reads were truncated from the first base where the Q score dropped below 3, and paired-end assembled with a minimum merge length of 200 reads. Merged reads with more than one base-pair error were discarded and the remaining demultiplexed. The bacteria and eukaryote data were concatenated into one file each (bacteria.fasta, eukaryote.fasta) and dereplicated. Unique sequences of bacterial 16S rRNA were compared to the Ribosomal Database Project (RDP; Cole et al., 2014), and unique sequences of eukaryotic 18S rRNA mapped against the Protist Ribosomal 2 database (PR2; Guillou et al., 2013) for chimera detection. Prior to clustering reads into operational taxonomic units (OTUs) with a 97% similarity, representative sequences were sorted by abundance so that the most important sequences would be at the start of the file and used as seed for the clustering. Taxonomy was assigned using the QIIME package (Caporaso et al., 2010) and the uclust assigner. By default, the program performs a search to record reference sequences matching a minimum of 90% of each query entry (OTU), and assign the most specific taxonomic label according to a minimum consensus fraction (0.51 by default) of the matches. This approach enabled taxonomic assignment of poorly represented taxa in reference databases. Operational taxonomic units with no match were discarded.

Rarefaction curves were generated in QIIME using the Chao1 and Shannon metrics (Fig. S1). This analysis demonstrated that approximately 2,500 reads per sample for both bacteria and eukaryote datasets were required to adequately capture the diversity within each sample. Samples with <2,500 reads (one for bacterial eDNA and one for eukaryote eDNA and eRNA samples) were discarded from subsequent analyses (Table S1). To enable robust comparisons between eDNA and eRNA datasets, an equivalent number of samples were kept per station for each type of molecule, resulting in the additional removal of one bacterial eRNA sample (Table S1).

For the remaining data manipulation and analysis, three datasets were used for both bacteria and eukaryotes: (1) containing all OTUs (referred to as ‘untrimmed’); (2) only containing OTUs with at least two unique sequence reads present within the whole dataset (referred to as ‘trimmed by singletons’); and (3) including only the OTUs found in both DNA and RNA compartments from each sample (referred to as ‘trimmed by shared OTUs’).

Data analysis and statistics

Samples were grouped into near-field (≤250 m of the FPSO) and far-field (>250 m) stations from each side of the FPSO. To visualize the effect of trimming by singletons and by shared OTUs, the number of reads and OTUs in each dataset were compared. The OTUs removed from the shared OTUs trimming method were further investigated by displaying bar plots of the relative abundance of the ten most important classes for bacteria and phyla for eukaryotes using the R package ‘phyloseq’ (McMurdie & Holmes, 2013), and by performing a Mann–Whitney U test with 999 permutations on these taxa to detect significant abundance differences between the near- and far-field stations.

Prior to further analysis, read counts were normalized among samples. The large discrepancy in read abundance among samples (up to five times) prevented the use of typical normalization techniques such as variance stabilization (R package DESeq2; Love, Huber & Anders, 2014) or cumulative sum scaling (R package metagenomeSeq; Paulson et al., 2013), and a subsampling-based normalization strategy was employed following the method described in Aguirre de Carcer et al. (2011). Briefly, multiple rarefactions (100 times) based on the median number of reads among all samples were performed, followed by a rarefaction to the lowest number of reads among all samples. Since a large number of singletons may represent PCR or sequencing artefacts (Majaneva et al., 2015; Brown et al., 2015), these were removed prior to statistical analysis.

Using normalized data kept after the two different trimming techniques, observed OTUs and the Shannon diversity index were analyzed for each sample. To test for the effect of the FPSO on the bacterial and eukaryotic alpha-diversity of eDNA/eRNA datasets, a nonparametric t-test using 999 permutations was performed between near and far-field stations. To determine whether the eDNA or eRNA dataset contained the most abundant number of OTUs significantly affected by the FPSO activity, a non-parametric t-test with 999 permutations was conducted on OTUs present in at least three samples, between the near- and far-field station groups. Differences in beta-diversity between and within the near- and far-field stations were tested using an Adonis and PERMDISP tests with 999 permutations on Bray-Curtis distance matrices, using square root transformed data. Non-metric multidimensional scaling plots were obtained using the R package ‘phyloseq’ (McMurdie & Holmes, 2013). Correlations between bacterial and eukaryotic eDNA/eRNA OTUs and macro-fauna beta-diversities, and with environmental variables were investigated using a Mantel test based on 999 permutations. The tests were performed on square root transformed data using Bray-Curtis distance matrices for biological data, and an Euclidean distance matrix for environmental data. For the later, only variables showing a significant change between the near-field and far-field stations were used (i.e., barium and AFDW; Johnston, Elvines & Newton, 2015), and square root transformed prior to the distance matrix computation. Except for the bar plots and nMDS, all analyses mentioned above were computed using the QIIME pipeline (Caporaso et al., 2010).

Results

High-throughput sequencing output

Each of the 18 sediment samples collected in this study were co-extracted for both eDNA and eRNA and PCR-amplified for total bacteria and eukaryotes, resulting in the total production of 72 PCR-amplicons (36 for bacteria and 36 for eukaryotes; Table S1). The no template PCR controls remained negative in all PCR trials. All 72 amplicons were sequenced on an Illumina™ MiSeq instrument, yielding a total of 2,134,982 and 1,040,624 eDNA and eRNA sequence reads, respectively. The water blank negative controls (ddH2O sample) contained a total of 524 raw reads (130 for bacteria and 494 for eukaryotes) in the eDNA library, and 405 raw reads (234 for bacteria and 171 for eukaryotes) in the eRNA library, of which none remained after quality filtering. The positive controls were assigned at 99.9% to the three target taxa. After removal of unmatched and low-quality reads, a total of 883,155 reads and 33,995 OTUs were obtained for bacteria (eDNA: 25,100; eRNA: 22,211), and a total of 768,374 reads and 11,854 OTUs (eDNA: 7,045; eRNA: 8,059) for eukaryotes. Removal of samples with insufficient diversity coverage led to the removal of one bacterial eDNA and eRNA sample (FE250b), and one eukaryotic eDNA (FE250b) and eRNA (FE250c) sample (Table S1).

Effect of trimming methodologies

For the bacterial dataset, the removal of singletons (i.e., OTUs present only once across the entire eDNA/eRNA datasets) resulted in a 1% loss of total reads and 37% loss of OTUs (Table 1). Similar results were observed for eukaryotes where 0.4% of reads and 29% of OTUs were removed. Trimming OTUs that did not occur in both eDNA/eRNA amplicons of the same sample resulted in a total reduction in OTUs of 78%, and total read reduction of 27%. For eukaryote data, trimming by shared OTUs had a larger impact on OTU and read removal, discarding an additional 51% of OTUs for a total reduction of 80%, and reducing the total read abundance by 28%. The mean number of reads (counts) per removed OTUs was of 3.1 and 8.6 for bacteria and eukaryotes respectively.

Table 1 Number of operational taxonomic units (OTUs) and reads per dataset.

Brackets show portion of reads removed by each technique from the raw dataset.

	Trimming technique	Reads	OTUs	
		Total	eDNA dataset	eRNA dataset	Total	eDNA dataset	eRNA dataset	
Bacteria	Raw data	87,1557	52,9180	34,2377	33,746	25,114	21,840	
Trimmed by singletons	85,8919 (1%)	52,2630	33,6289	21,108 (37%)	18,564	15,752	
Trimmed by shared OTUs	63,3643 (27%)	34,8860	28,4783	7,505 (78%)	7,505	7,505	
Eukaryotes	Raw data	76,7855	41,4271	35,3584	11,844	7,025	8,058	
Trimmed by singletons	76,4392 (0.4%)	41,2625	35,1767	8,381 (29%)	5,379	6,241	
Trimmed by shared OTUs	55,3651 (28%)	30,8285	24,5366	2,317 (80%)	2,317	2,317	
Notes.

SD Standard deviation

eDNA environmental DNA, and

eRNA environmental RNA

Figure 2 summarizes the total number of bacterial and eukaryotic OTUs that were either shared between both eDNA and eRNA data of the whole datasets, shared between eDNA and eRNA from the same sample, or those that were restricted to one dataset type following the two distinct data treatments. The percentage of shared OTUs between the eDNA and eRNA bacterial datasets increased from 22% to 36% after singleton removal (Table 1). For eukaryotes, removing singletons improved the percentage of shared OTUs in the eDNA and eRNA datasets from 20 to 28%. The percentage of unshared OTUs were highest in the eDNA dataset for bacteria, while in the eukaryotic dataset most of the unshared OTUs were in the eRNA.

Figure 2 Venn diagrams displaying shared and unshared operational taxonomic units (OTUs) among environmental DNA (eDNA) and RNA (eRNA) amplicons of the bacteria (A, B) and eukaryote (C, D) datasets.

Untrimmed data are represented in (A) and (C) datasets, with the removed singletons shown in (B) and (D)).

After trimming by singletons, the bacterial datasets (kept OTUs) were mainly composed of Gammaproteobacteria (eDNA 57.2%, eRNA 77.9%), Deltaproteobacteria (eDNA 12%, eRNA 8%) and Alphaproteobacteria (eDNA 8.5%, eRNA 4.8%; Table S3). Metazoa composed the large majority of eukaryote OTUs (eDNA 84.6%, eRNA 85.5%), followed by Dinophyta (6.2%) and Radiolaria (2.8%) in eDNA, and by Cercozoa (7.4%) and Cilliophora (3.5%) in eRNA (Table S3).

Trimming by shared OTUs slightly changed the relative abundance of each taxa of the kept data, but did not change their decreasing order (Table S4 ). The majority of bacterial OTUs removed from trimming by shared OTUs were Proteobacteria belonging to Deltaproteobacteria (eDNA 19%, eRNA 30.4%) and Gammaproteobacteria (eDNA 22.7%, eRNA 28.2%; Fig. 3, Table S3). In eDNA, the taxonomic groups Planctomycetia (9.6%), Alphaproteobacteria (7.1%) and Verrucomicrobiae (5.7%) also accounted for a substantial part of the trimmed OTUs. Most of the eukaryote OTUs removed were Metazoa (eDNA 60.5%, eRNA 42.8%), followed by Dinophyta (13.9%) and Apicomplexa (10.1%) in eDNA, and by Ciliophora (19.7%) and Cercozoa (19.7%) in eRNA (Table S3). Among all these groups, only eDNA Verrucomicrobiae (T-stat = 2.39, P-value = 0.03; Table S6) and eRNA Planctomycetia (T-stat = −2.78, P-value = 0.03; Table S6) expressed a significant relation with distance from the platform, both being more abundant in the near-field stations.

Figure 3 Relative abundance of the ten most important bacterial classes (A) and eukaryotic phyla (B) removed during the trimmed by shared OTUs method.

Abundance values for each class/phylum are stacked in order from greatest to least, separated by a thin horizontal line. eDNA, environmental DNA; eRNA, environmental RNA.

Comparison of bacteria and eukaryotes eDNA and eRNA datasets

Alpha-diversity

Non-parametric t-tests on alpha-diversity metrics of the eDNA dataset trimmed by singletons (kept OTUs) showed that there was no significant difference between the near- and far-field stations for bacteria or eukaryotes (Table 2). The same was observed with the equivalent eRNA bacterial data, and eRNA eukaryote data. When using the ‘trimmed by shared OTUs’ method, significant differences between the near- and far-field stations for the observed OTUs metric were observed with bacterial eDNA (p-value = <0.01) and with eukaryotic eDNA (p-value = 0.04), and for the Shannon index with bacterial eDNA (p-value = <0.01).

Table 2 Non-parametric t-tests on alpha-diversity metrics observed Operational Taxonomic Units (OTUs) and Shannon index between near field stations (≤250 m) and far field stations (>250 m), on the data kept after the different trimming methodologies.

Datasets	Observed OTUs	Shannon index	
			Near-field	Far-field	t-stat	p-value	Near-field	Far-field	t-stat	p-value	
Bacteria	Trimmed by singletons	eDNA	2,306	2,348	−0.33	0.71	9.6	9.83	−1.07	0.32	
eRNA	1,274	1,495	−2.08	0.06	7.86	8.05	−0.82	0.43	
Trimmed by shared OTUs	eDNA	807	1,356	−6.24	<0.01	7.56	8.63	−5.04	<0.01	
eRNA	958	1,111	−1.57	0.13	7.25	7.42	−0.79	0.44	
Eukaryotes	Trimmed by singletons	eDNA	550	572	−0.30	0.77	6.75	7.06	−0.77	0.48	
eRNA	686	830	−1.91	0.07	7.05	7.47	−1.05	0.29	
Trimmed by shared OTUs	eDNA	277	415	−2.16	0.04	5.78	6.45	−1.62	0.13	
eRNA	326	432	−1.42	0.17	5.67	6.42	−1.56	0.14	
Notes.

eDNA environmental DNA

eRNA environmental RNA

Significant p-values are in bold.

Significant OTUs, beta-diversity and correlation to environmental variables

The non-parametric t-test analysis identified OTUs significantly associated with the near- or far-field station groups, by comparing their frequencies across samples groups. For both taxa, these OTUs were more abundant in eDNA than eRNA (Table S7). Trimming by shared OTUs reduced the number of significant OTUs by 52 (eDNA) and 26 (eRNA) in bacteria, and by 99 (eDNA) and 86 (eRNA) in eukaryotes (Table S7).

Non-metric multidimensional scaling plots (Fig. 4) demonstrated that near- and far-field stations tended to cluster separately for both taxa. Our distance-based test for homogeneity of multivariate dispersion (Permdisp) indicated that communities within the near- and far-field stations groups were homogeneous in all datasets (Table S8). However, significant variance between replicates were observed for bacterial eDNA (p-value = 0.03) and eRNA (p-value = 0.04) data of the trimmed by singleton dataset, and for bacterial eDNA (p-value = 0.03) of the trimmed by shared OTUs dataset (Table S9).

Figure 4 Non-metric multidimensional scaling (nMDS) plots.

(A) Trimmed by singletons and (B) trimmed by shared operational taxonomic units (OTUs) data of both environmental DNA (eDNA) and RNA (eRNA) of bacteria and eukaryotes. NF, near-field; FF, far-field.

Table 3 Adonis and Mantel tests.

Analysis of the strength and statistical significance of sample groupings among datasets, and correlations between beta-diversity matrices of micro (bacteria and eukaryotes) and macro-fauna, and with distance matrix of selected environmental variables.

Datasets	Adonis	Mantel test	
			Near-field VS Far-field	Correlation with macro-fauna	Correlation with env. variables	Correlation DNA/RNA	
			R2	p-value	r	p-value	r	p-value	r	p-value	
Bacteria	Trimmed by singletons	eDNA	0.07	0.01	−0.53	<0.01	0.52	<0.01	−0.05	0.80	
eRNA	0.07	0.21	−0.06	0.78	0.04	0.82	
Trimmed by shared OTUs	eDNA	0.10	<0.01	−0.51	<0.01	0.58	<0.01	0.31	0.08	
eRNA	0.07	0.01	−0.08	0.73	0.08	0.63	
Eukaryotes	Trimmed by singletons	eDNA	0.08	0.04	−0.35	0.05	0.52	<0.01	0.47	0.02	
eRNA	0.12	<0.01	−0.14	0.48	0.24	0.09	
Trimmed by shared OTUs	eDNA	0.10	0.02	<0.01	0.99	0.32	0.02	0.89	<0.01	
eRNA	0.11	<0.01	−0.09	0.67	0.18	0.30	
Notes.

eDNA environmental DNA

eRNA environmental RNA

Significant p-values are in bold.

When comparing the correlation of bacterial eDNA/eRNA beta-diversity matrices between near- and far-field stations, eDNA provided stronger correlations and/or lower p-values (Table 3). This trend was also observed with correlations to the distance matrices of macro-fauna and environmental variables (Table 3). For eukaryotes however, this tendency was not always observed as eRNA showed slightly stronger correlations and more significant results between the near- and far-field stations, but weaker correlations with macro-faunal assemblages and with environmental variables. Overall, correlations between eDNA and eRNA beta-diversities were stronger for the trimmed by shared OTUs datasets, and only significantly positive among the eukaryote datasets (Table 3). For bacteria, the trimmed by singletons datasets demonstrated particularly weak correspondence (r =  − 0.05; p-value = 0.8) between eDNA and eRNA.

Discussion

The aims of this study were to assess whether high-throughput sequencing (HTS) data obtained from eDNA and/or eRNA provide the best proxy for evaluating environmental impacts and to explore how data processing steps may influence these results. We focused on benthic habitats at bathysmal depth, within close proximity to an offshore oil production platform.

High-throughput sequencing outputs

In our datasets, the number of bacterial eDNA OTUs (Table 1) was slightly higher than eRNA. Environmental DNA is more resistant to posthumous degradation, and may persist several years in marine sediment (Dell’Anno, 2005). Analysis of the eukaryote data showed a differing pattern with more eRNA OTUs compared to eDNA. The proportion of OTUs found only in eDNA or eRNA datasets varies substantially among previous investigations. For example, Dowle et al. (2015) found more bacterial eRNA than eDNA OTUs in their datasets, and Hu et al. (2016) and Laroche et al. (2016) obtained more protist eRNA than eDNA OTUs, while Pawlowski et al. (2014) and Pochon et al. (2015) detected more protist eDNA than eRNA OTUs. It is possible that many of the unshared eRNA OTUs represent PCR artefacts, especially from the reverse transcription of RNA to cDNA, and sequencing errors (Egge et al., 2013; Ficetola et al., 2015). The RNA conversion to complementary DNA (cDNA) requires the use of a reverse transcriptase which, as opposed to DNA polymerase, lack proof reading activity, creating point mutations in some of the cDNA sequences (Svarovskaia et al., 2003; Houseley & Tollervey, 2010). Reverse transcriptase also frequently ‘jump’ from one template to another during the transcription process. Known as template-switching, this action can lead to two different outcomes: (1) production of chimeric cDNA sequences from intermolecular template switching, and (2) creation of shortened isoform sequences from intramolecular template switching (Cocquet et al., 2006); the latter being more problematic as it is unlikely to be detected during bioinformatics analysis. Finally, the use of random hexamer primers for cDNA synthesis is known to introduce nucleotide bias at the beginning of the 5′-end of sequence reads (Hansen, Brenner & Dudoit, 2010). Because the pre-mentioned artefacts are subsequently augmented by the gene-specific PCR amplification, they would become integrated into the HTS library and potentially not be removed during bioinformatics filtering processes. The incertitude linked to potential intrinsic error rates from reverse transcription limits accurate comparison of eDNA and eRNA datasets. The use of a RNA control (e.g., synthetic oligomers) and technical (PCR) replicates could help identifying these artefacts and improve concordance between the eDNA and eRNA profiles. Trimming OTUs by their shared presence in eDNA and eRNA represents an alternative way of discarding reverse transcription artefacts.

In this study, bacterial and eukaryote RNAs were treated using the same protocol during the reverse transcription process. However, eukaryote cDNA required more PCR cycles (35 instead of 30 for bacteria) to amplify sufficient material, potentially producing more false transcripts and explaining the differences observed between the two taxonomic groups (Fig. 2). Another possibility could be the presence of rare but highly active eukaryote OTUs which were not detected in the eDNA dataset because of their low abundance. Particularly high activity and growth from some of the rare biota has been previously reported for both bacteria (Campbell et al., 2011; Hugoni et al., 2013) and microbial eukaryotes (Logares et al., 2014).

Effect of trimming methodologies

Trimming OTUs not simultaneously present in both eDNA and eRNA assemblages from the same samples resulted in twice as many OTUs being removed compared to discarding singletons only, and removed 27-fold (bacteria) and 70-fold (eukaryotes) more reads. However, further investigation suggested that it mostly resulted in the removal of legacy DNA, DNA deposited from the water column, and technical artefacts. This improved the sensitivity and accuracy of our environmental metabarcoding analysis (Tables 2 and 3) and enabled the detection of differences between the effects of the production platform on near- and far-field stations. In particular, it allowed us to disclose the divergence between beta-diversities of near-field and far-field stations, and improved the correlations between the beta-diversity and environmental variables distance matrices (Table 3).

Removal of single sequence detection (singletons) to reduce PCR and sequencing artefacts is a controversial practice. Although many of these detections are true errors (Parada, Needham & Fuhrman, 2016), some may constitute real OTU occurrences. In studies not particularly interested by the rare biota, the trade-off of producing false negatives to remove false positives is usually accepted (Majaneva et al., 2015; Bálint et al., 2016). Systematic inclusion or exclusion of singletons is arbitrary and not necessarily adapted to all situations. For example, the likelihood of retaining false positives increases with the sequencing depth of HTS libraries (Ficetola, Taberlet & Coissac, 2016). Some researchers have therefore suggested the use of a read detection threshold based on sequencing depth (Bálint et al., 2016). However, Ficetola, Taberlet & Coissac (2016) and Lahoz-Monfort, Guillera-Arroita & Tingley (2016) strongly discouraged systematic removal of taxa based on an abundance threshold, unless high detection probability of real OTUs and sufficient biological or technical replicates are available to identify false positives. Instead, they propose the application of site occupancy-detection modelling (SODM) methods which take into account false positive and negative detection probabilities, and which have proven to work well with few replicate samples (Schmidt et al., 2013; Lahoz-Monfort, Guillera-Arroita & Tingley, 2016).

In our study, we used co-extracted DNA and RNA to segregate real OTUs from artefacts. This approach allows for an informed elimination of artefactual OTUs while also discarding data from legacy DNA. However, trimming data by shared OTUs between co-extracted eDNA and eRNA material is particularly stringent and may result in the removal of a substantial number of false negatives. As an alternative, biological or technical replicates could be used in parallel to discriminate genuine and false detections. For example, biological replicates could be pooled together prior to trimming by shared OTUs, and subsequently demultiplexed to allow evaluation of within station variability. It is worth mentioning that the application of unsupervised oligotyping methodologies such as the one proposed by Eren et al. (2015), could further improve sensitivity of environmental metabarcoding monitoring studies by discriminating closely related taxa, and will likely gain in popularity as tools are being developed.

Operational taxonomic units not shared between both eDNA and eRNA datasets

Among the bacterial datasets, most of the removed OTUs (i.e., those not simultaneously found in both eDNA and eRNA) belonged to Gammaproteobacteria and Deltaproteobacteria, which also constituted the main classes of the singleton trimmed dataset. This suggests that most OTUs removed by the shared OTUs trimming method may have arisen from PCR or sequencing errors originating from highly abundant unique sequences which subsequently associated to these classes (Huse et al., 2010; Quince et al., 2011). Another possibility could be that these abundant taxa have more polymorphisms. In such cases, removal of these rare variants would produce false negatives. Nonetheless, their small contribution to weighted (abundance-based) analyses, as demonstrated in the present study, would unlikely affect results.

Most of the eukaryotes which were removed during the trimming procedure were Metazoa, Dinophyta and Apicomplexa in eDNA, and Ciliophora and Cercozoa in eRNA (Fig. 3, Table S5). These phyla are also the most abundant in the datasets trimmed by singletons (Table S3). None of these showed a significant correlation with distance from the platform (Table S6). Just as for bacteria, the important abundance of these phyla in the discarded eDNA/eRNA OTUs is likely a consequence of PCR/sequencing errors originating from the most abundant unique sequences. This is further supported by the fact that the decreasing order of relative abundance of the most important taxa is the same between the datasets of kept OTUs and removed OTUs resulting from the shared OTUs trimming.

Comparison between eDNA and eRNA

When performing alpha-diversity on the trimmed by singletons datasets, differences were observed between eDNA and eRNA, with the latter mainly showing stronger dissimilarities between the near-field and far-field stations. This is not surprising as eDNA can contain genetic information of past and current communities, inflating alpha-diversity and potentially masking the present effects of the platform (Corinaldesi, Danovaro & Dell’Anno, 2005; Dell’Anno, 2005).

Beta-diversity analysis of the singleton trimmed data showed more significant differences between the near- and far-field stations with bacterial eDNA, while only small differences could be observed between eukaryote eDNA and eRNA. Relationships between micro-eukaryotic and the macro-faunal assemblages and environmental variables were also much stronger than with eDNA. This suggest that while eRNA may represent a better proxy for assessing ongoing anthropogenic effects on species diversity, eDNA seems more reliable for assessing effects on community composition. This is especially true for bacteria where no correlation could be observed between eRNA data and macro-fauna, or with environmental variables (Table 2). These results contrast with other similar studies (Pawlowski et al., 2014; Visco et al., 2015; Dowle et al., 2015; Pochon et al., 2015; Laroche et al., 2016; Hu et al., 2016) where eRNA always out-performed eDNA when correlating microbial and macro-faunal assemblages, or when correlating these assemblages with environmental conditions. Nonetheless, these studies used a less stringent OTU filtering method prior to statistical analyses, removing only those not present in both eDNA/eRNA compartment of their whole dataset. Consequently, an important number of OTUs from legacy DNA may have subsisted in their datasets, decreasing the observed sensitivity to local conditions of the eDNA-derived communities. Also, the relationship between eRNA concentrations and activity does not always align. Differences such as life history, life strategies and non-growth activity between organisms play an important role in RNA production (Blazewicz et al., 2013). Ribosomal eRNA such as those targeted in this study (16S and 18S) can have a much wider range of expression (over 3 orders of magnitude) than ribosomal eDNA (ca. 1 order of magnitude; (Fegatella et al., 1998)). Fluctuation in DNA copy number between taxa and in RNA expression may account for the strong variability observed among the bacterial eDNA and eRNA replicates (Table S8). This variability increases confidence intervals, reduces statistical power, and increases the risk of not identifying impacts (Type II errors; Underwood, 1993).

For accurate assessment of false positive and negative OTU occurrences, and sufficient evaluation of within station biological variability, optimal number of replicate samples is often a compromise between available funds and time, and the desired level of precision and sensitivity. Parameters such as targeted taxa, the choice of primer set and the sequencing depth of HTS libraries all play an important part in taxa detection probability, which in turn, dictates the optimal replication level (Ficetola et al., 2015). In studies assessing whole communities, the number of replicates should ideally depend on the taxa with the lowest detection probability. Ficetola et al. (2015) give an example where a taxon with a 0.5 probability would normally require a minimum of eight replicates to ensure detection. For routine monitoring surveys of benthic communities however, where the non-detection of rare biota is unlikely to change the outcomes of studies, the replication level does not have to be as high. In such cases, the number of replicates may be more influenced by the level of heterogeneity found within stations.

An important consideration when using single-time sampling designs over a contamination gradient is that it assumes that natural factors equally influence communities among samples (Wiens & Parker, 1995). This bias can be particularly problematic with small and short-lived organisms like bacteria. In low impacted environments, such as those explored in this study, biological interactions may supplant the effect of environmental factors, increasing spatial variability (Schafer, 2000), and creating stronger discrepancies among biological replicates. In this situation, collecting more biological replicates per stations should be favoured over technical replicates, in order to mitigate the effect of micro-patchiness.

Conclusion

The resolution, sensitivity and accuracy of metabarcoding techniques used to measure anthropogenic impact on microbial communities can be substantially affected by whether eDNA or eRNA is used and by the choice of pre-analysis data treatment method. Disparities in results may also occur between eukaryote and bacterial eDNA/eRNA, that should be considered before analyzing environmental metabarcoding monitoring data. Our study showed that when OTUs are solely trimmed by singletons, eRNA represents a better proxy for alpha-diversity as it is more likely to contain living biota only. Nonetheless, read abundance information from eDNA data was, for bacteria, superior to its counterpart for assessing environmental impacts on beta-diversity. In weakly impacted environments, this may be even more important as eRNA concentration can be highly dependent and sensitive to temporal interactions from restricted local conditions that can be unrelated to the anthropogenic activity. For this reason, read abundance information from bacterial eDNA seems more suited for assessing the effect of oil and gas drilling and production operations, where closest monitored stations are located far from the pollution source (ca. 250 m). Overall, trimming OTUs not shared in both DNA and RNA compartments of each sample improved the sensitivity and accuracy of our results. While this is true for our study site, further investigations will be needed to evaluate the extent of these findings.

Supplemental Information

Figure S11 Rarefaction curves on Chao 1 and Shannon metrics for each of the 18 environmental DNA and RNA amplicons of bacteria (A) and eukaryotes (B) analyzed using high-throughput sequencing

Click here for additional data file.

Table S1 Samples identification, coordinates and environmental DNA/RNA amplification results

Click here for additional data file.

Table S2 Analytical methods used for the external laboratory analysis of the physico-chemical characterization of sediments. Adapted from Johnston et al. (2014) and Skilton et al. (2015)

Click here for additional data file.

Table S13 Read abundance and proportion of bacterial and eukaryotic environmental DNA (eDNA) and RNA (eRNA) per phylum, based on the trimmed by singletons datasets

Click here for additional data file.

Table S4 Read abundance and proportion of bacterial and eukaryotic environmental DNA (eDNA) and RNA (eRNA) per phylum, based on the datasets trimmed by shared operational taxonomic units (OTUs)

Click here for additional data file.

Table S5 Read abundance and proportion of bacterial and eukaryotic environmental DNA (eDNA) and RNA (eRNA) per phylum, based on operational taxonomic units (OTUs) removed from trimming by shared OTUs

Click here for additional data file.

Table S6 Mann-Whitney U test between the near-field ( ≤250 m) and far-field stations ( >250 m), based on the removed operational taxonomic units (OTUs) of the trimmed by shared OTUs datasets

eDNA, environmental DNA; eRNA, environmental RNA.

Click here for additional data file.

Table S7 Number of indicator operational taxonomic units (OTUs) per dataset associated with the near-field or far-field station groups

Click here for additional data file.

Table S8 Permdisp analysis assessing the beta-diversity variance within the near-field and far-field station groups

Click here for additional data file.

Table S9 Distance-based test for homogeneity of multivariate dispersions (Permdisp) analysis with 999 permutations, assessing the beta-diversity variance between biological replicates

Deviations are from centroids. Significant p-values are in bold.

Click here for additional data file.

We thank Olivia Johnston and Deanna Elvines (Cawthron) for sample collection, and OMV New Zealand Limited for providing access to samples and corresponding environmental data.

Additional Information and Declarations

Competing Interests

Author Contributions

DNA Deposition

Data Availability

Dr. Xavier Pochon is an Academic Editor for PeerJ.

Olivier Laroche conceived and designed the experiments, performed the experiments, analyzed the data, contributed reagents/materials/analysis tools, wrote the paper, prepared figures and/or tables, reviewed drafts of the paper.

Susanna A. Wood conceived and designed the experiments, contributed reagents/materials/analysis tools, wrote the paper, reviewed drafts of the paper.

Louis A. Tremblay, Gavin Lear and Joanne I. Ellis reviewed drafts of the paper.

Xavier Pochon conceived and designed the experiments, contributed reagents/materials/analysis tools, wrote the paper, prepared figures and/or tables, reviewed drafts of the paper.

The following information was supplied regarding the deposition of DNA sequences:

The raw sequencing reads are publicly available in the Sequence Read Archive (SRA) under the accession numbers SRR5215546 to SRR5215617.

The following information was supplied regarding data availability:

The raw sequencing reads are publicly available in the Sequence Read Archive (SRA) of the GenBank database under the accession numbers SRR5215546 to SRR5215617 of project number PRJNA369002.

Alternatively, the raw sequencing reads can also be accessed in figshare (doi:10.6084/m9.figshare.4600174):

https://figshare.com/articles/Laroche_et_al_2017_Maari/4600174

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
