# Peer review of "Metabarcoding monitoring analysis: the pros and cons of using co-extracted environmental DNA and RNA data to assess offshore oil production impacts on benthic communities"

_PeerJ, doi:10.7717/peerj.3347_

## Round 0.1 · original submission · Major Revisions

I have heard back from two reviewers with contrasting opinions. While both agree the paper is well written, reviewer 2 has concerns about the bioinformatic analyses that at least require more explanation and at most new analyses and a restructuring of the paper; hence my decision is major revisions. I look forward to receiving a new version of this paper.

Reviewer 1 ·

Basic reporting

This paper by Laroche et al is another excellent contribution by the Cawthron in the area of environmental DNA. In many ways it is a scoping study but one that tackles some fundamental questions of marine metabarcoding on infauna. I have a number of minor modifications and/or suggestions below which I hope will assist the authors in re-submitting the manuscript.

The manuscript is well structured, the references suitable and the data analysed well. Overall the submission tests a key hypothesis about the concordance between eDNA and eRNA. It will be a welcome addition to the literature.

Experimental design

The experimental design in this manuscript is solid. In studies such as these it is always difficult to know a priori what levels of replication are required. There are two areas in the experimental design that I would like the authors to elaborate on in their revision (i) the experimental controls in the workflows and (ii) comment about the level of replication used in this study – be more explicit about the within and between site variability

In my general comments below there are a few additional notes regarding experimental design that need addressing .

Validity of the findings

When a paper breaks new ground such as this there are a number of outstanding questions, however, the authors have done a solid job of analysing a complex RNA/DNA dataset. As one last caveat it may be prudent to mention that the use of multiple PCR assays may improve the concordance between eRNA and eDNA profiles.

Additional comments

In the abstract Laroche et al state: Most studies only assess eDNA which, unlike eRNA, can persist in the environment after cell death
This statement is not accurate – RNA has been isolated from a number of substrates long dead (e.g. ancient maize) . the statement need to be revised to talk about the relative stability. Alternatively, the authors should speak to the unlikelihood of RNA surviving long in marine environments.

Abstract: ‘Therefore, DNA data do not necessarily reflect recent environmental change leading some researchers to advocate for the use of eRNA as an additional, or perhaps superior proxy for portraying ecological changes’ this sentence need to be re-written – I think the authours are trying to say that RNA may provide a more immediate census of the environment (than DNA) due to its relative stability.
Abstract “Based on these findings, we recommend that metabarcoding-based biomonitoring studies should, whenever applicable, be assessed using both eDNA and eRNA material” I think this statement need moderating – stating that there are some benefits to both and this is the ideal scenario, but that ultimately costs, specialised sample collections (to preserve RNA) and simplicity of workflows may need to be factored.
Line 86: replace ‘better’ with a more fitting descriptor.

Line 174 – Some more information is needed on the cDNA synthesis, it is unclear if the author used hexamers, targeted primers etc. Latter in the manuscript hexamers are discussed but I could not find it in the methods.
Line 187 – it is unclear if PCR replicates were conducted across all 72 amplicons?
The Authors have submitted the raw data to SRA but what about the filtered set? Perhaps consider putting the data up on Data Dryad? This is important if someone wanted to test (or reanalyze) the dataset.
Figure 1b is quite a strange layout – almost an empty box I would advise a re-think of this figure.

Figure 2 – It would be good to put eRNA and eDNA above the venn diagrams. I realize this is shown in the legend but it would be easier to read with ‘tags’ above the appropriate Venn.

There is not much information in the paper about controls. What turned up in the experimental controls?

Line 367 – it may be good to advocate for an RNA control (e.g. synthetic oligo) to help controls for RT-error. It could well be that the ‘simple’ removal of singletons is not sufficient in eRNA datasets. I am not proposing any further experiments but it seems that until the intrinsic error rates are determined that there will be severe difficulties in accurate comparisons of eDNA/eRNA datasets.

Line 400-405. While I don’t disagree with this hypothesis it could also be that populations of these abundant bacteria simply have more polymorphisms. It is a difficult problem to address, but simply removing singletons may be a blunt way to filter for artifacts especially for reads closely related to highly abundant OTUs. Some discussion on appropriate cut-offs to maximise signal:noise might be appropriate here.

Line 411, are there any other bacteria consistent with human waste? This is a pretty speculative stuff as there are a number of locations that ‘human’ bacterial can originate from (including contamination).
Line 446. There are also a lot of differences in copy number and this could easily influence the proportions and composition of eDNA/eRNA datasets like this.

Discussion. It would have been nice to see some discussion regarding the levels of replication and experimental design as this manuscript really paves the way for future metabarcoding applications like this. Were triplicates necessary? Would more sampling points and less replicates be more useful?

Discussion; This is an excellent paper in that, for the first time, it explores the value of eDNA/eRNA in tandem on infauna sediment. I think the authors have missed a bit of a trick in that there is space in the discussion to suggest the way forward with this kind of work. For example (i) replication needed (ii) numbers of sampling points (iii) DNA first then onto RNA due to cost? (iv) using multiple assays in combination. I guess I would liked to have seen a little more discussion of ‘what next’.

Reviewer 2 ·

Basic reporting

This manuscript aims to address an important scientific question, mainly the use of eDNA and eRNA in environmental assessment of benthic sediment habitats proximate to offshore oil production sites. However, unlike other published eDNA/eRNA studies, the present manuscript suffers from fatal flaws in terms of sampling design, data processing, and conclusions drawn from the underlying data.

From an organizational perspective, the manuscript is appropriately structured in terms of manuscript sections, figures, and tables. The writing is clear and does not require further proofreading for language or grammar.

For the most part the authors have cited a robust body of literature focused on combined eDNA/eRNA studies. However, I found references in other areas to be a bit lacking (e.g. justification of bioinformatics choices, supporting evidence for conclusions), and some citations provided to support speculation that in my view was not supported by the data on hand (e.g. Metazoan OTUs chalked up to extracellular DNA).

Experimental design

The experimental design is extremely limiting given the overarching questions driving this study - in my view, it fails to include appropriate checkpoints and controls that would enable robust assessment of the variables and environmental factors which may be impacting the observed microbial community patterns.

My major concerns regarding experimental design are as follows:

- Only 6 sites (for a total of 18 samples) were assessed during this study. This does not give nearly enough statistical power in terms of drawing ecological conclusions relevant for seabed monitoring; currently across the field, other benthic studies are assessing upwards of a hundred samples (sometimes thousands) per study.

- A Van Veen grab was used to collect samples, which is a fairly harsh sampling method (compared to box cores or multi/mega-corers) since it disrupts the sediment-water interface and would bias the recovery of benthic organisms. If the goal is to assess oil production impacts, a more sensitive sampling method would be preferable (e.g. multi-cores, push cores from ROVs) in order to capture as much of the benthic community as possible.

- DNA was only extracted from 2g of sediment (for a total of 36g of sediment extracted during the entire study). This would be OK (and standard practice) for assessing bacterial communities, but it is not likely to do a good job recovering a broad swath of the benthic eukaryotic community, especially meiofauna or macrofauna where a larger volume of sediment is needed to accurately capture the community (e.g. which the authors acknowledge, in their morphological methods where a larger volume of sediment is sieved for macrofauna).

- The parallel morphological methods (macrofauna identification) are not really comparable to the eDNA work, at least for eukaryotes - the methods in this study would only do a good job of assessing protist taxa in the 2g of sediment, and eDNA samples do not assess a large enough sample volume to capture the benthic meiofauna or macrofauna community fractions. Furthermore, the macrofaunal morphology is hardly discussed in the results and discussion, and so I am not sure why this was included in the first place.

- The authors did not include any negative controls or blank samples to assess the potential contribution of the microbes in PCR/sequencing kits and reagents. Blank samples are becoming standard practice - given the variable eRNA signal and stringent OTU filtering strategy employed here, I suspect the authors may actually be enriching for kit contaminant taxa (but there is no way to assess this because of the lack of blank/control samples - see Salter et al. 2014 https://bmcbiol.biomedcentral.com/articles/10.1186/s12915-014-0087-z)

- The bioinformatics methods may be removing real biological signal from the dataset. For example, on Line 217 the authors are using BLAST to assign taxonomy to eDNA/eRNA OTUs, requiring 97% similarity in their matches. Subsequently, “Operational Taxonomic Units with no match were discarded” - this is a very misguided strategy because 1) marine benthic environments would not be well represented in the databases, so most taxa are likely to have low matches (<95%) to the database, and 2) OTUs with high match percentages (>97%) have a higher likelihood of belonging to lab contaminants, which are likely to be well represented in public databases. Here, the authors could filter out kit control OTUs if they had included blank samples, but this is not possible given the study design.

- Given the above concerns, I cannot be confident in the downstream informatics comparisons (untrimmed vs. trimmed by singletons vs. trimmed by shared OTUs). These comparisons may be rather arbitrary given the lack of blank control samples and the lack of in-depth exploration of any other data patterns (e.g. looking at low-level taxonomy across families, genera, etc. - at the moment everything is summarized at a very high level, and it is difficult to relate the microbial community patterns to actual biological or ecological phenomena).

Validity of the findings

This manuscript does not hold up as a methods-focused paper. That does not mean the underlying eDNA dataset is not valuable, it just means the study design is not rigorous enough to test the two stated aims of using eDNA/eRNA to assess environmental impacts and exploring the effect of data processing.

For example, a large part of the discussion section is spent talking about the errors and problems associated with RNA data - but then the authors go on to conclude that their eRNA data shows the most significant and convincing patterns (without linking these potential patterns to contamination, errors, etc. introduced because of their study design).

There are other issues with the supporting evidence provided in the discussion - many observed patterns are spuriously linked to “PCR or sequencing errors” without any data provided to support this fact (e.g. line 404 - not all OTUs removed via bioinformatics filtering would be erroneous sequences; abundance-based filtering will arbitrarily remove biologically valid OTUs. The authors did not rigorously look for errors in their data via their bioinformatics workflow, at least as far as I can tell). In addition, metazoan OTUs are attributed to extracellular material (line 418) without acknowledging the likely presence of diverse meiofauna and protist specimens that would likely be present in the sediment samples.

I would suggest completely overhauling this manuscript to focus on the biological patterns - looking deeply at taxonomic patterns and framing this study in a more ecological context is the only way to overcome the current shortcomings in methodology and study design. However, even with such an overhaul the authors would have to seriously revisit their bioinformatics workflow and parameters to be more rigorous and in line with other recent studies in the field. For example, one interesting pattern worth exploring further would be the separation of Near Field vs. Far Field sample sites (consistent across bacteria and eukaryote eDNA PCoAs in Figure 4). What specific taxa are driving this pattern?

Additional comments

Other minor comments:

- Line 181: How were these primer sets chosen? Do they represent standard metabarcoding regions (e.g. Earth Microbiome Project recommended primers?)

- Line 183: What is the amplicon length of 16S and 18S amplicons?

- Line 324 - There is no such thing as “near significant” p-values. p=0.06 and p=0.07 are not significant

- Line 326 - Please report p-values here in the text, since you state they are significant.

---

## Round 0.2 · accepted · Accept

Both reviewers state that the new version of your work is well-revised, and I feel it is now acceptable to be published.

One note: Please ensure that the following comment from reviewer 2 is dealt with at the Proofs stage if not earlier:

Line 174: "All PCR amplifications included negative controls (no template samples) to ensure amplification of uncontaminated products." --> I suggest changing to "..to check amplification of..." ("ensure" seems like the wrong word here).

I look forward to seeing the online published version.

Reviewer 1 ·

Basic reporting

As reported in my previous review the quality of writing, figures and structure is of a high standard. The need for this kind of work is clearly articulated in a set of hypotheses.

Experimental design

As previously noted this is an initial foray into examining eDNA/eRNA communities in the context of oil/gas operations - the preliminary nature of this particular study is now well articulated in the manuscript.

The methods are now well described (with mention of controls) and data is now available for download at an external website.

Validity of the findings

As with the earlier draft I consider the data to be robust and appropriately analysed for the stated aim of the study. The authors duly acknowledge that more could (and should) be done when applying this in oil/gas studies when exploring baselines/impacts.

Additional comments

In the revised manuscript Laroche et al have offered up some edits to address the reviewers concerns. I enjoyed re-reading this paper and the result have certainly given me insights into eDNA/eRNA data we are planning on generating in our lab.

In my view the manuscript is ready for publication and think that the authors have done a solid job in addressing the concerns of the reviewers.

After reading the comments of the other reviewer and the authors response I would like to offer up a few additional comments – namely that the field of eDNA and eRNA is far from stable. There is not currently a set of ‘best-practice’ guidelines on how to generate, filter and present metabarcoding data such as this. In contrast, the literature offers up a set of (often) contrasting opinions on various aspects of metabarcoding. With the field ‘in flux’ it is somewhat easy to ‘throw stones’ at any study for perceived failures in sample design and data analytics. The salient point here is that the raw and processed data is available to others who think they can do a better job at analysing it. This is exacerbated by the fact that the field is moving at a pace that data generated a year earlier may be perceived to have been inappropriately generated/analysed. If the eDNA/eRNA field is to mature we need to see data published in a timely manner where we can iteratively learn from the approaches employed.

My perspective on the presented dataset differs from that of the other reviewer in that I think it adds to only a small number of studies that have looked at eDNA/eRNA data in tandem. Sure, the data are not perfect, the level of replicates is small and more could have been done in terms of technical replicates. However, this form of 20-20 hindsight is not overly constructive especially on a set of samples that is logistically difficult to collect. The authors have clearly articulated the preliminary nature of this work and inserted the appropriate caveats where relevant. This type of work is likely to become best-practice in monitoring of oil and gas impacts and as such is a valuable stepping-stone. Taken together it is the opinion of this reviewer that the study firmly meets the criteria for publication.

Reviewer 2 ·

Basic reporting

The authors have sufficiently responded to my previous comments - there was detail lacking in a number of key areas in the original manuscript, which the authors have now rectified. The information provided on negative controls and details of bioinformatics parameters has significantly strengthened and clarified the manuscript, and allayed the majority of previous concerns.

Experimental design

I agree that marine sediment sampling is not always easy, and I also appreciate the authors expanding on their limitations and constraints during sampling (particularly in regard to government/agency restrictions, and putting their sampling strategy in context with other comparable studies, and emphasizing that statistical assessment indicated that the sampling strategy employed here was adequate). Being up front about this information is particularly important for readers who are not necessarily familiar with marine systems.

Validity of the findings

The authors have stated that they are preparing a separate manuscript on the biological conclusions from this dataset, and this clarifies my comments regarding the scope and recommendations made in the manuscript. Given the additional text and methods added to the revised manuscript, the findings are much better supported now, given the dataset and bioinformatics workflow employed here.

Additional comments

One minor comment on wording:

Line 174: "All PCR amplifications included negative controls (no template samples) to ensure amplification of uncontaminated products." --> I suggest changing to "..to check amplification of..." ("ensure" seems like the wrong word here)